# Relationship between the Content of β-D-Glucans and Infection with *Fusarium* Pathogens in Oat (*Avena sativa* L.) Plants

**DOI:** 10.3390/plants9121776

**Published:** 2020-12-15

**Authors:** Michaela Havrlentová, Veronika Gregusová, Svetlana Šliková, Peter Nemeček, Martina Hudcovicová, Dominika Kuzmová

**Affiliations:** 1Department of Biotechnologies, Faculty of Natural Sciences, University of Ss. Cyril and Methodius in Trnava, 917 01 Trnava, Slovakia; gregusova4@ucm.sk (V.G.); kuzmicka@gmail.com (D.K.); 2National Agricultural and Food Centre, Research Institute of Plant Production in Piešťany, 921 68 Piešťany, Slovakia; svetlana.slikova@nppc.sk (S.Š.); martina.hudcovicova@nppc.sk (M.H.); 3Department of Chemistry, Faculty of Natural Sciences, University of Ss. Cyril and Methodius in Trnava, 917 01 Trnava, Slovakia; peter.nemecek@ucm.sk

**Keywords:** oat, β-D-glucans, artificial inoculation, *Fusarium*, DON, plant protection

## Abstract

In human nutrition, oats (*Avena sativa* L.) are mainly used for their dietary fiber, β-D-glucans and protein content. The content of β-D-glucans in oat grain is 2–7% and is influenced by genetic and/or environmental factors. High levels of this cell walls polysaccharide are observed in naked grains of cultivated oat. It the work, the relationship between the content of β-D-glucans in oat grain and the infection with *Fusarium graminearum* (*FG*) and *Fusarium culmorum* (*FC*) was analyzed. The hypothesis was that oats with higher content of β-D-glucans are better protected and the manifestation of artificial inoculation with *Fusarium* strains is weaker. In the 22 oat samples analyzed, the content of β-D-glucans was 0.71–5.06%. In controls, the average content was 2.15% for hulled and 3.25% for naked grains of cultivated oats. After the infection, a decrease was observed in all, naked, hulled and wild oats. As an evidence of lower rate of infection, statistically significant lower percentage of pathogen DNA (0.39%) and less deoxynivalenol (DON) mycotoxin (*FC* infection 10.66 mg/kg and *FG* 4.92 mg/kg) were observed in naked grains compared to hulled where the level of pathogen DNA was 2.09% and the average DON level was 21.95 mg/kg (*FC*) and 5.52 mg/kg (*FG*).

## 1. Introduction

Oats (*Avena sativa* L.) have received considerable attention for their high content of functional substances such as dietary fibers, phytochemicals and other substances with high nutritional value [1]. Essential amino acids, unsaturated fatty acids, total dietary fiber and especially the polysaccharide (1-3),(1-4)-β-D-glucan (hereafter referred as β-D-glucans) are the main compounds of the grain with high nutritional value and functionality [2]. It is reported that the interest in oat production has increased since the US Food and Drug Administration has recognized oat husks in diet reducing the risk of heart diseases due to the physiological effects of β-D-glucans [3]. Quality parameters of cereal grain is generally influenced by a number of factors such as agronomic inputs, the environmental cultivation conditions, as well as the plant status and plant health, especially. 

Fungal infections caused by different *Fusarium* species constitute one of the most serious problems for oat plants health. *Fusarium culmorum* (*FC*) and *Fusarium graminearum* (*FG*) belong to main causal agents of Fusarium head blight (FHB) [4], whether it is well known that oat cultivars are highly pathogen-resistant, do not require high cultivation inputs [5] and high disease resistance predetermines oat cultivars on organic farms [6]. In some cases, infections caused by *Fusarium* pathogens can reduce the overall yield and thus can cause economic losses [7]. Also, caused by the infection, grains can be contaminated with mycotoxins produced by fungi hosting on cultural plants [8]. In cereals, *Fusarium* pathogenic fungi negatively affect grain production and quality due to the contamination of grain with trichotecene mycotoxins [9]. The most toxicologically important *Fusarium* mycotoxins are deoxynivalenol (DON), T-2 toxin (T2), HT-2 toxin (HT2), zearalenone (ZEN) and fumonisin B1 (FB1) [10] whose consumption in high doses can cause nausea, diarrhea and vomiting in humans [11], while feed refusal, also vomiting and diarrhea are common symptoms for animals [10]. T2 and HT2 toxins are immune-toxic and potent inhibitors of protein synthesis [12]. The effects of *Fusarium* pathogens are negative regarding the grain quality, cause damage during grain storage, negatively affect the content and quality of aleurone layer proteins (albumins and globulins), as well as active responses of plant to these pathogens (α-amylase inhibitors, xylanase inhibitors, serpins, etc.) [13]. The European Commission has set limits on *Fusarium* mycotoxins in crops feeding the world. DON and ZEN for unprocessed cereals, intermediate products (e.g., flour) and finished products for human consumption are limited to the maximum recommended levels of 8000 μg/kg and 2000 μg/kg, respectively [14]. A recommendation for monitoring the levels of T2 and HT2 toxins in cereals has been adopted by the European Commission with benchmark levels for the levels of T2 and HT2 toxins 1000 μg/kg in unprocessed oats and 200 μg/kg in processed oats, including bran and flakes directly used for consumption, respectively [15].

Cell walls are dynamic structures that represent key determinants of overall plant form, plant growth and development and the plant responses to environmental and pathogen-induced stresses [16]. Plants exposed to physical, chemical or biological stress factors must be able to react by cell surface hardening and related tolerance or resistance [17]. Plant cell wall polysaccharides, including β-D-glucans, determine responses to various environmental factors at specific stages of plant development [18]. Rapid cell wall reinforcement can reduce success of pathogen’s penetration. Such enhancements occur due to the accumulation of β-(1-3)-glucan [19,20], β-D-glucans [21] and proteins in the area between the cell wall and the cell membrane [22]. The gel-layer formed by β-D-glucans in the cell wall may also act as a defensive wall protecting the cell from invasion by fungi but also providing a potential signal system to indicate when such an attack is under the way [23].

Oat mature grains are good natural sources of β-D-glucans [24]. These cell wall polysaccharides occur only in selected plant species, especially grasses well adapted to different and often much stressed environments [21]. Wild types of cultivated plant species are good natural sources of interesting genes determining interesting features such as nutritional quality of the grain or resistance to pathogens and other kinds of biotic and abiotic stress factors [6,22]. Therefore, besides cultured *A. sativa* also wild species of *Avena* genus were analyzed in our work. In the study the aim was to investigate the role of β-D-glucans in plant protection by analyzing the relation between the content of β-D-glucans naturally occurred in the oat cell wall and the manifestation of infection caused by the artificial fungal inoculation with *FC* and *FG*. 

## 2. Results

In all *Avena* spp. (*A*) samples analyzed, the content of β-D-glucans ranged between 0.71% (*A. fatua*) and 5.06% (naked *A. sativa* PS-215, breeding line). In *A. sativa* genotypes, the content of monitored polysaccharide ranged between 0.96% (hulled *A. sativa* Plato) and 5.06% (naked *A. sativa* PS-215). In hulled *A. sativa*, the content of β-D-glucans ranged between 0.96% and 3.90% with an average amount of 2.15%. In naked *A. sativa*, the content of monitored of the metabolite ranged between 1.61% and 5.06% with an average of 3.25%. In naked oats, higher content of β-D-glucans was observed in our experiment compared to hulled ones. Among the other *A.* species (wild types), the lowest content of β-D-glucans was found in the grains of *A. fatua* (0.71%) and the highest was in *A. strigosa* (2.55%). More than 2.00% of β-D-glucans were observed in the grains of *A. byzantina* (2.52%). In grains of *A. ludoviciana*, *A. canariensis* and *A. murphyi,* the average amounts of the monitored polysaccharide were 1.67%, 1.30% and 1.29%, respectively (Table 1).

In our experiment, in control oat grains without the inoculation was the average content of β-D-glucans 1.87%. In oat samples inoculated with *FC* and *FG* a decrease (−4% and −22%, respectively) in the content of the polysaccharide was observed (1.79% and 1.46% were the mean values of β-D-glucans after the FC and FG infection, respectively) compared to the control. In naked samples of *A. sativa*, the average content of β-D-glucans in the control was 3.25% and it decreased rapidly after the fungal infection to 1.92% after the artificial inoculation with *FC* (−41%) and to 1.62% after the infection with *FG* (−51%). The average β-D-glucans content in control samples of hulled oats was 2.15% and after the infection with *FC* it decreased to 1.34% (−38%) and after the infection with FG to 1.29% (−40%) (Table 1). Paired two sample t-test proved statistically significant (*n* = 22; *p* < 0.05) decrease in the β-D-glucans levels after both types of infection compared to the levels in control samples. This decrease achieved the significance *p* = 0.017 for the *FC* infection and even higher significance, *p* = 0.001, for *FG*.

In naked oat samples with significantly higher amount of β-D-glucans both, statistically lower presence of the pathogen DNA (0.39%) and lower content of the mycotoxin DON (10.66 mg/kg for *FC* infection, 4.92 mg/kg for *FG* infection) were detected compared to hulled samples (Table 1). In hulled oat samples, the proportion of the pathogen DNA was 2.09%. The mycotoxin DON was observed in hulled oat grains after the artificial inoculation with *FC* in the average amount 24.18 mg/kg and after the *FG* infection it was 4.82 mg/kg. Comparing the differences in the manifestation of artificial inoculation in oat grains it is possible to state that the fungus *FC* had more negative effect on oat plants compared to *FG*.

Statistical evaluation showed that in naked oats was the artificial inoculation less noticeable compared to hulled ones. Genotypes with naturally occurring higher content of β-D-glucans in the grain were more moderate infected by *FC* and showed lower levels of DON toxin (4.08 mg/kg and 2.32 mg/kg of DON, respectively) as well as lower levels of pathogen DNA (0.02% and 0.04%, respectively) (Table 1). On the other hand, genotypes with lower amount of β-D-glucans achieved higher DON levels in the grain as well as higher level of pathogen DNA content after the artificial inoculation. Similar results as with the FC infection were obtained also after the infection with *FG* (Table 1). In naked oat genotypes, lower presence of pathogen DNA and DON were observed compared to hulled oat samples, suggesting that naked oats showed less infection compared to hulled ones. The DON content in naked oat genotypes was 4.92 mg/kg on average and 5.52 mg/kg in hulled oats. Also, the pathogen DNA content was lower in naked oat genotypes (0.39%) than in hulled ones (2.09%). Identically to the *FC* infection, it can be concluded that after the *FG* infection naked oat genotypes with higher amounts of β-D-glucans showed less infection compared to hulled oat samples with lower amounts of the polysaccharide in the grain. According to our results it is also shown that the infection of *FG* was more moderate than that one of *FC*.

The differences in tested parameters: the β-D-glucans content in the control, the β-D-glucans content after the infection, the content of DON and the percentage of the pathogen DNA were evaluated by ANOVA and post hoc test (LSD) for following oat genotypes categories: naked, hulled and wild. For the *FC* infection were found statistically significant (*n* = 22; α < 0.05) differences in the DON accumulation between naked and hulled oat genotypes (*p* = 0.014), whereby the average DON content in naked oats was 10.66 mg/kg and in hulled oats it was 21.95 mg/kg. Similar differences were found also in the presence of pathogen DNA in naked and hulled oats (*p* = 0.001) and hulled and wild (*p* = 0.005). Also correlation analysis proved significant negative correlation between DON and β-D-glucans in the controls (*r* = −0.430; *p* = 0.046). For the *FG* infection ANOVA results found statistically significant (*n* = 22; α < 0.05) differences in the levels of β-D-glucans in control samples between naked and wild genotypes (*p* = 0.050). In the case of pathogen DNA percentage, similar results were observed between naked and hulled genotypes (*p* = 0.001) and between hulled and wild genotypes (*p* = 0.050). In the case of FG any significant correlations among tested parameters were detected. 

Our results indicate that naked oat genotypes artificially inoculated with *FC* responded better to the artificial inoculation and expressed less infection than hulled ones. Statistically significant differences in the presence of infected pathogen DNA after the artificial inoculation with *FG* were also shown in naked and hulled oats. The ANOVA with LSD test showed more detailed statistically significant differences in the percentage of pathogen DNA after the infection with *FG* among naked grains of *A. sativa*, hulled grains of *A. sativa* and wild types. The average content of pathogen DNA was 0.39% for naked oats after the infection with *FG*, 2.09% for hulled oats and 0.48% for wild types of *A.* These results indicate that naked oat genotypes infected with FG disposed of the lowest content of pathogen DNA compared to other samples.

Similarities as well as differences between studied genotypes from the measured parameters point of view are depicted in dendrogram form cluster analysis (Figure 1 and Figure 2). For both infections (*FC* and *FG*), genotypes were grouped into four clusters based on their β-D-glucans content in the control, β-D-glucans content after the infection, the DON content and the percentage of pathogen DNA. 

Four groups of *A.* can be observed in the Figure 1 based on their response to *FG* artificial inoculation. In the group of “red” genotypes (*A. strigosa*, SV-5 and PS-223 with relatively high content of β-D-glucans in control samples compared to other samples) the β-D-glucans content was increasing after the *FG* infection, which could be caused by the activation of natural plant protection tools based on the passive mechanism of the cell wall. Higher content of β-D-glucans in the grain before the infection could be one of factors responsible for this activation. Generally, a decrease in the β-D-glucans content in most of the analyzed samples was observed after the fungal infection, which could be due to the fungus using the polysaccharide β-D-glucans as an energy source when infecting and penetrating into the plant. In contrast to this global trend, in this “red” group an increase in the content of the polysaccharide after the infection was detected. The “green” group (*A. byzantina*, Norik, Vaclav, PS-222, Dunajec, PS-215, PS-218 and PS-219) is characterized with high content of β-D-glucans (>3.00% besides *A. byzantina* with 2.52%) in the control but after the infection no increase in the content of the polysaccharide was observed. The third, “blue” group of oats (*A. ludoviciana*, Vojtech, Dalyup, Vit, *A. murphyi*, Hronec, *A. fatua*, Numbit and Plato) did not dispose of high content of β-D-glucans in control samples and genotypes showed high content of both, pathogen DNA and DON after the infection with *FG*. Važec and *A. canariensis* represent the fourth “orange” group where grains dispose of low content of β-D-glucans in the control with a decreasing trend after the infection and beside relatively low pathogen DNA they accumulate the highest content of DON.

Four groups of oats genotypes can be also observed in the Figure 2 based on their response to the artificial inoculation with *FC*. In the “red” group of oats (Hronec, Vaclav, Važec and *A. strigosa*) the content of β-D-glucans was relatively high (>2.20%) in the control and increased after the infection with *FC*. The DON accumulation was middle to low as well as the presence of pathogen DNA was not very high. The “green” group of oat genotypes (*A. byzantina*, SV-5, PS-222, Norik, PS-219, Dunajec, PS-215, PS-218, *A. canariensis*, *A. fatua*, *A. murphyi* and PS-223) was characterized by relatively high content of β-D-glucans in the control and lower content of pathogen DNA. The “blue” group of genotypes (*A. ludoviciana*, Vit, Dalyup and Vojtech) disposed of low content of β-D-glucans in the control with a decreasing trend after the infection and strong infection symptoms represented by high content of DON and high percentage of pathogen DNA. The fourth “orange” group (Numbit and Plato) represents oat genotypes with low content of β-D-glucans which increased after the infection. The contents of DON and pathogen DNA were the highest in this group compared to other groups in the cluster analyses.

In general, green colored genotypes represent an average group of plants with middle properties and infection manifestation. Blue group contains genotypes with high content of the pathogen DNA and slightly higher levels of DON which indicates higher sensitivity of these genotypes to the *Fusarium* infection. Orange group represents the smallest group of genotypes with high percentage of the pathogen DNA and extremely high levels of the mycotoxin DON which indicates that these genotypes are extremely sensitive to the fungal infection and to the accumulation of the mycotoxin DON. The last, red colored group is represented by genotypes with high levels of β-D-glucans in both, control and infected samples. Cluster analysis revealed different distribution pattern of genotypes based on the type of infection. In both infections the naked *A. sativa* genotypes (except PS-223 in the case *FC*) clustered with the blue group of genotypes represented by high content of the pathogen DNA as well as relatively high levels of the DON. 

The main advantage of principal component analysis (PCA) is a possibility of depicting measured properties and samples (genotypes) as well as the multidimensional space reduction into 2 or 3 dimensions. Both biplots (Figure 3 and Figure 4) reflect results of previous cluster and correlation analysis. For *FG* can be the first principal component interpreted as an axle of the β-D-glucans content before (control) and after the infection and partly divides genotypes according their infection manifestation (content of the pathogen DNA and the content of DON). The second biplot reveals different relationships and is more straightforward. The first principal component indicates state of infection based on the percentage of pathogen DNA and the DON content, where the blue and orange groups of genotypes have mostly positive PC 1 values. The second principal component (PC 2) is related with the β-D-glucans content before (control) and after the infection. 

Results in Figure 3 suggest that there is a negative relationship between the content of β-D-glucans, the cell wall polysaccharide in oat grain and the manifestation of the infection caused by *FG*, that is, that the presence of higher β-D-glucans content in oat grain could evoke lower infection and thus higher protection of the plant against the infection with the *FG*. Another pair of vectors, β-D-glucans content after the *FG* infection and concentration level of DON provides the same opposite relationship. Genotypes with relatively high β-D-glucans levels dispose with lower DON concentrations. A negative correlation can be identified also between the β-D-glucans content in the control and the percentage of pathogen DNA (*r* = −0.3702; *p* < 0.001) and between the percentage of pathogen DNA and the β-D-glucans content after the infection with *FC* (*r* = −0.2268; *p* < 0.001) (Figure 4). The highest negative correlation (*r*= −0.4297; *p* < 0.001) was recorded between the content of β-D-glucans in the control and the content of DON after the infection with *FC*. 

## 3. Discussion

The average content of the cell walls metabolite β-D-glucans in our experiment ranged between 0.71% (*A. fatua*) and 5.06% (naked *A. sativa* PS-215), where the mean value was 1.87%. Havrlentová and Kraic [24] determined the content of β-D-glucans in oat mature grain at values from 2% to 6%, with an average of 4%. Their results correspond to the determined average values of β-D-glucans in European oat genotypes (3.9% and 3.6%, respectively) published by Redaelli et al. [25]. The β-D-glucans content of individual oat samples depends not only on the genotype but also on the environment in which oat plants are cultivated (soil, climatic conditions and agrotechnology). Zute et al. [26] claimed that naked oats contain greater amount of β-D-glucans compared to hulled oats. Also in our work it was detected that the average content of β-D-glucans was higher in naked *A. sativa* genotypes (3.25%) compared to hulled ones (2.15%). Havrlentová et al. [27] published also that hulled oat genotypes differed statistically significant from the naked ones with higher content of total dietary fiber and crude fiber in the grain and oppositely, naked oats were characterized by higher content of β-D-glucans and total proteins. This fact is also confirmed by Tiwari and Cummins [28], who stated that naked oat grains are nutritionally better than traditional hulled ones. The absence of an indigestible surface layer can be responsible for higher metabolic energy. Wild types of plants are very often good natural sources of interesting genes and their products useful for plant breeding [29]. More than 2.00% of monitored β-D-glucans was in our research indicated in *A. strigosa* (2.55%) and *A. byzantina* (2.52%). These plants can be used in the next research as genetic resources for production of higher amounts of β-D-glucans, metabolite with high nutritional and functional value [2,30].

The plant cell wall is one of factors responsible for plant reaction to abiotic and biotic stress factors [16] by cell surface hardening [17]. Rapid cell wall reinforcement can reduce the success of pathogen to penetrate and this cell wall thickening can be done by β-D-glucans [21,23]. Initial observations have suggested that oat genotypes disposing of higher β-D-glucans content are more resistant to pathogen manifestations caused by parasitic organisms being “inhibited” in their pathogen manifestations by thicker cell wall in which β-D-glucans are produced in higher amounts [19]. In our previous work it was proven that β-D-glucans in the oat grain can protect the grain against high temperature [22]. It this work, it was expected that oat genotypes with higher β-D-glucans content may be more resistant to artificial inoculation with *FC* and *FG*. Results and statistical evaluations obtained in this experiment show that higher content of β-D-glucans assured plants more resistant to pathogenic manifestation. In our experiment, grains with higher content of β-D-glucans in the control produced plants accumulating lower DON levels in the grain as well as less pathogen DNA after an artificial inoculation with two *Fusarium* strains. 

Martin et al. [31] observed similar results with barley samples after the infection with *FG*. Genotypes with the highest β-D-glucans content displayed the lowest DON content. However, the principle of the interaction between the content of β-D-glucans and contamination with DON as well as the role of this metabolite in the plant resistance is a source of discussion. On one hand, higher β-D-glucans content might enhance natural resistance of the plant against fungal pathogen by contributing to type V resistance (resistance against toxins accumulation) caused by antioxidant activity of β-D-glucans [31,32]. Indeed, several studies demonstrated the inhibitory potential of natural antioxidant compounds of plant origin on the production of mycotoxins [33,34,35]. On the other hand, it is documented that β-D-glucans are able to bind several *Fusarium* toxins in vitro [36,37]. 

In our experiment, a decrease in the content of β-D-glucans was observed in most of samples after the artificial inoculation. This can have two possible explanations. One is the fact that parasitic organisms (in our case fungi of *Fusarium*) use oat β-D-glucans as a source of energy for their nutrition in the hosting plant. Glucose units are a very good and easy digestible source of energy for various evolving and growing organisms. Lower β-D-glucans content in infected grains might be attributed on one hand to the activity of fungal β-glucanases and other cell-wall degrading enzymes of *Fusarium* pathogens [38,39,40,41]. On the other hand, the reduced supply of sugars and other nutrients can lead to a reduced synthesis of β-D-glucans in the developing grain [42,43,44]. Our results demonstrate that *FG* colonizing the grain and synthesizing mycotoxins caused reduction of the grain size and shape (data not shown) as well as reducing the content in β-D-glucans. This plant pathogen causes a plant disease characterized by mycelial development and cell disintegration (and possibly degradation) [45] and probably also breakdown of β-D-glucans as a structural component of the cell wall. 

Differences in the responses of the plant to pathogens used in the experiment were detected in our work. In genotypes artificially inoculated with *FC* the content of β-D-glucans was about 0.33% higher than with *FG*. Differences were also observed in the accumulation of DON in the grain. The DON content after the *FC* infection was in all analyzed samples higher compared to the infection with *FG*. These observations indicate that the infectant *FC* acts more aggressively on oat plant compared to *FG*. It is evident that differences in the reaction of plants to artificial inoculation can be related to different ability of *Fusarium* isolates to produce DON as well as their aggression. It is documented in the literature that in wheat (*Triticum aestivum* L.), is the resistance against different *Fusarium* strains approximately identical [46,47]. However, in the case of oat it is still not clear enough, what is the behavior of the infection with *Fusarium* [48], although the influence of the interaction genotype and isolate is evident. Besides, the accumulation of toxins, *Fusarium poae* (*FP*) and *Fusarium langsethiae* (*FL*) altered the β-D-glucans content in oat grains. Depending on the genotype and environmental conditions, the β-D-glucans contents increased or decreased subsequent to *FP* and *FL* infections [49]. In particular, the β-D-glucans content in grains of the naked variety Samuel substantially increased following infections by both *FP* and *FL* (+48% and +64%, respectively), whereas the changes in β-D-glucans content caused by FHB were limited for the other genotypes (approximately +20%). Hence, no clear tendency in β-D-glucans content was identified [49]. In a very similar study on barley grains, a 10% reduction of β-D-glucans content was observed in six barley varieties after inoculation with *FG* [31]. On one hand, the reduction of β-D-glucans content might be attributed to the β-glucanase activity observed from *Fusarium* pathogens [38]. On the other hand, since β-D-glucans are mainly concentrated in the outer layers of oat grains [50], such variations in β-D-glucans content might be attributed to changes in grain morphology caused by the infection. Hence, tracing the content of β-D-glucans in the oat grain after the infection will help to understand the impact of the infection not only on the biochemical composition of the cereal grain but also on the plant metabolism, its health status and overall quality. The cultivation of resistant varieties is the most sustainable, cheaper and effective way to control yield [29] and contamination of primary food sources with mycotoxins. 

## 4. Materials and Methods 

### 4.1. Plant Material

Twenty-two samples of different species and genotypes of oats were used in the experiment. All genetic resources of oats were provided for research purposes by the breeder and curator of genetic resources of oats in the Slovak Republic, Peter Hozlár, PhD. (National Agricultural and Food Centre—Research Institute of Plant Production, Research and Breeding Station at Vígľaš-Pstruša, Detva). Wild species *A. byzantina* C. Koch (Genome ACD), *A. canariensis* Baum. (Ac), *A. fatua* L. (ACD), *A. ludoviciana* Durieu. (ACD), *A. murphy* Ladiz. (AC) and *A. strigose* Schreb. (As), as well as cultivated *A. sativa* L. (ACD) samples were selected for this study. Among *A. sativa*, hulled genotypes Norik, Vaclav, Vit, Dalyup, Vojtech, Numbit and Plato and naked Hronec, Važec, Dunajec, SV-5, PS-223, PS-215, PS-218, PS-222 and PS-219 were used. Seeds of oats were seeded in pots in three parallel replications. One repetition was used as a control, without the artificial inoculation Two repetitions were artificially inoculated. From each genotype, 15 panicles inoculated with *FC*, 15 ones inoculated with *FG* and 15 panicles of the control variant were collected. After harvest, mature grains were husked by hand and stored at −4 °C for further analysis.

### 4.2. Artificial Inoculation 

For the artificial inoculation, the spraying method combined with polyethylene bag coverage was used [51,52]. From each sample the panicles were artificially inoculated in the flowering phase using a spraying method with inoculum of both, *FC* and *FG*. After inoculum application, panicles were covered with polyethylene bags for 48 h. 

Selected fungal isolates of *FC* and *FG* obtained from tested cereal species were used to prepare the inoculum. The *Fusarium* isolates were collected from commercial wheat fields from naturally infected wheat spikes collected in different regions of the Slovak Republic and are maintained in the microorganism’s collection of the Research Institute of Plant Production (Piešťany, Slovak Republic). The fungal colonies of *FC* and *FG* were grown on potato dextrose agar plates (PDA, Difco) or synthetic nutrient agar plates (SNA) at 25 °C for 21 days in dark [53]. The inoculum was prepared by the surface flooding of agar plates with sterile distilled water from 90 mm diameter Petri dishes and scraping the sporulated aerial mycelium with a loop. Concentration of inoculum was measured with a hemocytometer and adjusted to approximately 5 × 10^6^ propagules per ml. Approximately 1 mL of the conidial suspension was applied to each oat panicle. 

### 4.3. Determination of β-D-Glucans

Total content of β-D-glucans in analyzed samples was determined using Mixed-linkage β-glucan assay kit (Megazyme, Bray, Ireland) based on the method published by McCleary and Codd [54]. This method is accepted by the Association of Official Agricultural Chemists (AOAC) (Method 995.16) and the American Association for Clinical Chemistry (AACC) (Method 32–23). Mature grains were milled and passed through 0.5 mm sieve using an Ultracentrifugal Mill (ZM 100, Retch, Germany). Samples were suspended and hydrated in a sodium phosphate buffer (pH 6.5), incubated with purified lichenase enzyme and an aliquot of filtrate was reacted with purified β-glucosidase enzyme. The glucose product was assayed using an oxidase/peroxidase reagent. The values of the β-D-glucans were achieved for each sample as the mean according to three replications and dry matter as a percentage.

### 4.4. Quantification of DON 

A commercial competitive enzyme-linked immunosorbent assay (Ridascreen Fast DON; RBiopharm, Darmstadt, Germany) was used to determine the DON concentration in oat samples with the limit of detection <0.2 mg/kg (ppm) and the limit of quantification 0.2 mg/kg (ppm)/oats: 0.36 mg/kg (ppm). The method is accepted by the AOAC according to the Performance tested method program and FGIS (Federal grain inspection services) program of the Grain Inspection, Packers and Stockyards Administration of the United States Department of Agriculture (USDA/GIPSA). Mature oat grains were milled and passed through 1 mm sieve using Ultracentrifugal Mill (ZM 100, Retch, Germany) and stored at −20 °C until analyses. Samples were watered with distilled water, stirred vigorously for three minutes and filtered using Whatman No.1. The amount of 50 µL of the filtrate was used per well in the test. The absorbance (of the wells) was determined photometrically at 450 nm (MRX II. Dynex Technologies, Chantilly, VA, USA). Microplate reader (Dynex Technologies, Chantilly, VA, USA) equipped with Revelation^®^ software was used for the quantification of DON. Samples with high contamination were diluted in distilled water according to the manufacturer’s instructions.

### 4.5. Extraction and Quantification of DNA

Mycelia of *FC* and *FG* grown on Petri dishes were scraped off and homogenized to a fine powder using liquid nitrogen. Pathogen DNA was extracted using DNeasy Plant Maxi Kit (Qiagen, Germany). Grains of analyzed oats (infected and controlled) were gently milled and 1 g of the powder was used to isolate total DNA using the same DNeasy Plant Maxi Kit as for fungal mycelia. Control of quality and concentration of isolated DNA was performed using a NanoDrop1000 Spectrophotometer (Thermo Fisher Scientific Inc., Waltham, MA, USA). All DNA samples obtained from oats were diluted to the concentration 25 ng/µL.

Quantification of pathogen DNA in oat grains was performed by real-time PCR using an ABI PRISM^®^ 7000 (Applied Biosystems, Thermo Fisher Scientific Inc., Waltham, MA, USA) in MicroAmp 96-well optical plates (Applied Biosystems, Thermo Fisher Scientific Inc., Waltham, MA, USA). The TaqManTM probe *FC92s1* 5′ FAM 3′ MGB—AAAGAAGTTGCAATGTTAGTG and primer pair *FC92s1* forward TTCACTAGATCGTCCGGCAG and *FC92s1* reverse GAGCCCTCCAAGCGAGAAG were used and the obtained amplicon size was 92 bp specific for *F. culmorum* by Reference [55]. For the quantification of *F. graminearum* DNA the TaqManTM probe *Tri5* 5′ FAM 3′ MGB—AACAAGGCTGCCCACCACTTTGCTCAGCCT and primer pair *Tri5* forward TCTTAACACTAGCGTGCGCCTTCT and *Tri5* reverse CATGCCAACGATTGTTTGGAGGGA were used and the obtained amplicon size was 193 bp by Reference [56]. Reaction mixture in a volume of 25 µL consisted of 12.5 µL TaqMan Universal PCR Master Mix (Applied Biosystems, Thermo Fisher Scientific Inc., Waltham, MA, USA), 300 nM forward and reverse primers, 200 nM TaqManTM probe and 1 µL of DNA (25 ng). PCR conditions were as follows: 95 °C for 10 min, 40 cycles: 95 °C for 15 s, 60 °C for 1 min. Five standard dilutions of *FC* and *FG* DNA (1; 10; 100; 1000 and 10,000 pg/μL) were used as standard for standard curves in three runs per cycle. ABI PRISM^®^ 7000 software (Applied Biosystems, Thermo Fisher Scientific Inc., Waltham, MA, USA) was used to evaluate the results, in which unknown samples were quantified from measured Ct values by interpolation using a standard curve derived regression equation. The DNA content of *FC* and *FG* was expressed in ng of pathogen DNA per g of grains and the percentage of infection was calculated as (*Fusarium* DNA/total DNA) × 100.

### 4.6. Statistical Evaluation

Results obtained from chemical analyses were statistically analyzed using the JMP 11.0 software. Following statistical methods were used: correlation analysis (CA), analysis of variance (ANOVA) with Post Hoc test (LSD) and principal component analysis (PCA).

## 5. Conclusions

In oat genotypes of cultivated *A. sativa* and wild types an artificial inoculation with two isolates of *Fusarium* (*F. culmorum*, *FC* and *F. graminearum*, *FG*) was performed. Plants showed different reactions to the infection. Naked oats generally disposed of higher content of β-D-glucans both in the control variant and also after the inoculation compared to the hulled ones. In naked oats, the average β-D-glucans content in control samples was 3.25% and it decreased about 42% after the inoculation with *FC* and about 51% after the inoculation with *FG*. In hulled oats, β-D-glucans content in control samples was in the average 2.15% and after the inoculation it decreased about 38% (*FC*) and 40% (*FG*), respectively. As an evidence of lower rate of infection, statistically significant (*p* < 0.001) lower percentage of pathogen DNA (0.39%) and lower content of the mycotoxin DON (10.66 mg/kg for *FC* infection and 4.92 mg/kg for *FG*) in naked samples were detected. In hulled oats, the proportion of pathogen DNA after the artificial inoculation with *Fusarium* isolates was 2.09%, DON was in the mean level of 21.95 mg/kg for *FC* and 5.52 mg/kg for *FG*. According to the results and statistical evaluation, naked oats better reacted to the artificial inoculation by accumulating lower levels of both, DON and pathogen DNA. This can be due to protective character of β-D-glucans in the cell wall. According to our results, *FC* had statistically significant higher negative effect on oat grain compared to *FG*. The cultivation of oat genotypes with high β-D-glucans content can be recommended to reduce the risk of mycotoxin contaminated grains and to further promote the production of health promoting foods.

## Figures and Tables

**Figure 1 plants-09-01776-f001:**
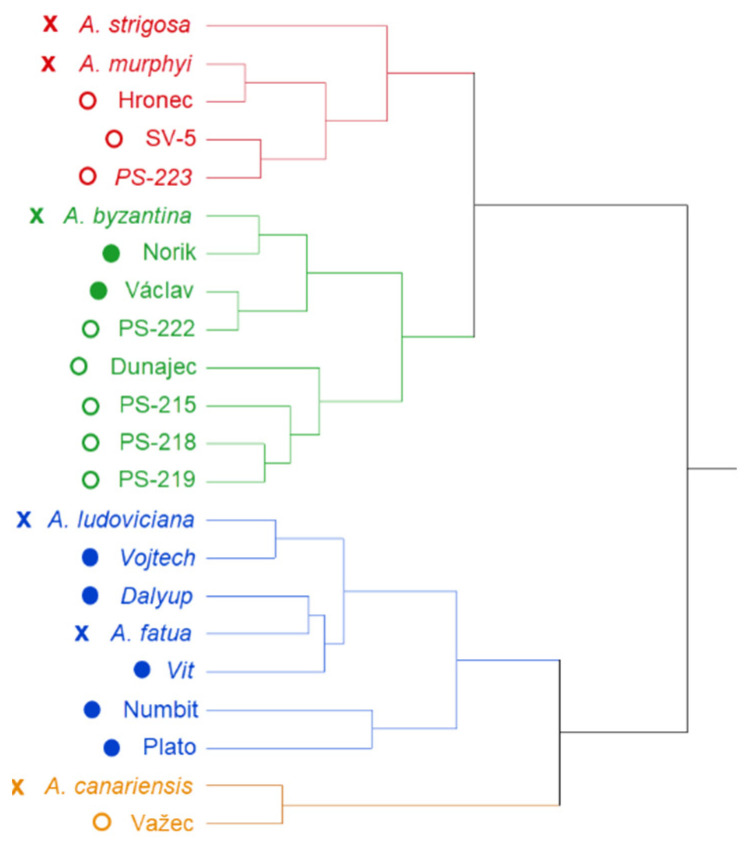
Grouping of analyzed oat genotypes after the infection with *F. graminearum* using a hierarchical cluster analysis. Types of markers indicate naked (○), hulled (●) and wild (x) oat genotypes.

**Figure 2 plants-09-01776-f002:**
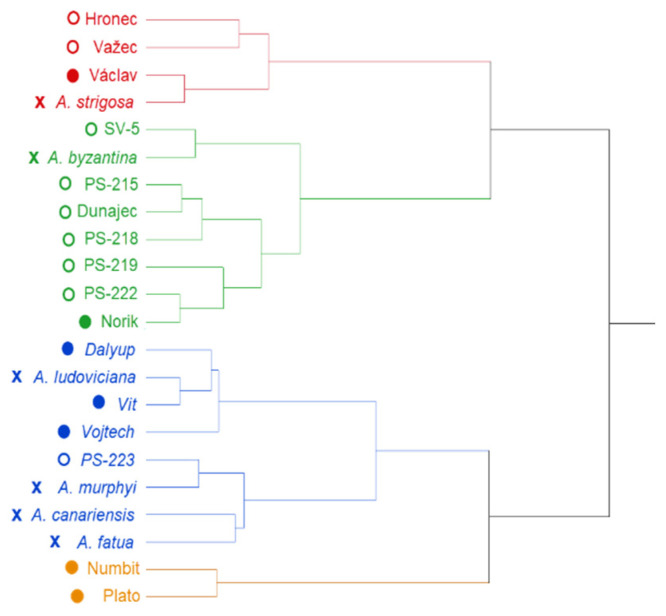
Grouping of analyzed oat genotypes after the infection with *F. culmorum* using a hierarchical cluster analysis. Types of markers indicate naked (○), hulled (●) and wild (x) oat genotypes.

**Figure 3 plants-09-01776-f003:**
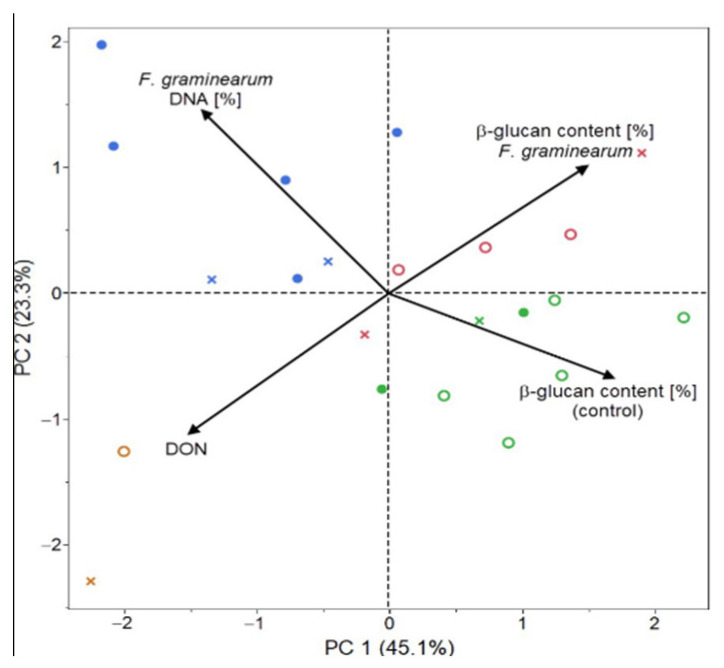
Principal Component Analysis (PCA) of analyzed parameters in 22 oat genotypes after the infection with *FG*.

**Figure 4 plants-09-01776-f004:**
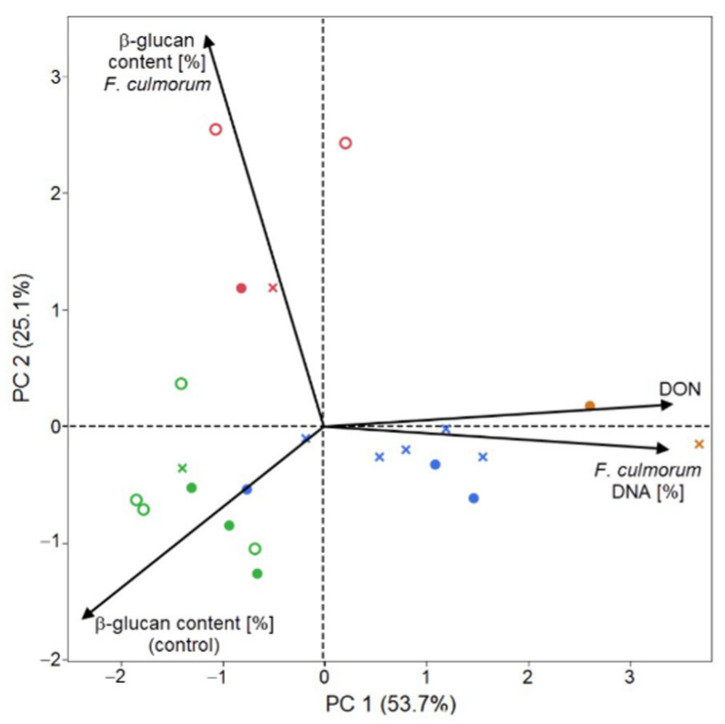
Principal Component Analysis (PCA) of analyzed parameters in 22 oat genotypes after the infection with *FC*.

**Table 1 plants-09-01776-t001:** Content of the parameters in the analyzed oat samples inoculated with *Fusarium* strains.

No.	Sample		β-D-Glucans in Control (%)	*FC* Inoculation			*FG* Inoculation		
				β-D-Glucans after Inoculation (%)	DON (mg/kg)	Pathogen DNA (%)	β-D-Glucans after Inoculation (%)	DON (mg/kg)	Pathogen DNA (%)
1.	*A. strigosa*		2.55	3.30	19.39	0.28	2.90	3.17	0.28
2.	*A. ludoviciana*		1.67	1.40	28.19	1.23	1.64	6.52	1.20
3.	*A. byzantina*		2.52	1.41	2.32	0.04	1.23	1.16	0.04
4.	*A. fatua*		0.71	0.89	15.29	1.23	0.93	4.78	1.23
5.	*A. canariensis*		1.30	0.89	23.33	0.03	0.72	14.93	0.03
6.	*A. murphyi*		1.29	1.21	14.14	0.08	1.33	4.63	0.08
7.	*A. sativa*/hulled	Vaclav	3.90	3.25	15.43	0.59	0.94	4.59	0.59
8.		Vojtech	1.54	0.78	22.33	2.16	1.89	7.51	2.16
9.		Dalyup	2.77	1.48	25.96	1.98	1.9	5.32	1.98
10.		Numbit	1.16	1.55	29.20	3.55	1.68	10.47	3.55
11.		Vit	1.62	1.14	28.84	1.71	1.63	1.22	1.73
12.		Plato	0.96	1.11	37.20	4.29	0.94	3.22	4.29
13.		Norik	3.39	1.13	10.27	0.32	1.48	2.12	0.33
14.	*A. sativa*/naked	Hronec	1.90	4.60	7.47	0.23	1.55	3.24	0.23
15.		Važec	3.15	3.94	21.68	0.92	1.00	12.77	0.92
16.		SV-5	2.54	2.20	4.33	0.11	2.22	2.84	0.12
17.		PS-223	1.61	0.88	5.93	0.07	1.89	2.76	0.08
18.		PS-222	3.73	0.77	13.77	0.46	0.99	3.72	0.46
19.		Dunajec	4.98	2.00	14.20	0.44	1.63	8.34	0.41
20.		PS-215	5.06	1.84	9.70	0.11	2.10	2.94	0.10
21.		PS-218	4.43	1.75	4.39	0.18	1.54	3.85	0.17
22.		PS-219	4.72	1.36	15.25	1.30	1.64	3.24	1.30

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
