# Peer review of "Relationship between the Content of β-D-Glucans and Infection with Fusarium Pathogens in Oat (Avena sativa L.) Plants"

_plants, 2020, doi:10.3390/plants9121776_

Round 1

Reviewer 1 Report

The research is interesting, well-conducted, and well-written. However, I have one major concern about the experimental design: the three replicates of the experiment were not established at the same time, i.e. two replicates (inoculated plants) were grown in a first step, and another replicate (non-inoculated plants) were grown later in a second step. This is not correct, because the plants were not cultivated in parallel. You even mentioned that B-glucan content may be affected by the environment (L255-257). I believe that the research is worth publishing with minor revisions, but frankly speaking, this concern is problematic for stating about sound results, and I think that it cannot be solved now.
Other minor comments are reported here below:
L56: ZEN
L74-77: this is clear to me, however, I believe that it should be re-written to be more clear and easy-to-read. I would formulate a scientific hypothesis to be tested. Also, at L334 you mentioned that genotypes were selected upon the B-glucans content. Indeed, it is not clear if B-glucan content is previously known information or it was determined in this work. Please clarify in M&Ms and the description of the aim of this article.
L 117: the term "pathogenic DNA" is not correct! You might want to write "pathogen DNA" or "Fusarium DNA" or "fungal DNA" or FC DNA", etc.
L76: you could cite these fungal species earlier in the text when the toxins are described. In the Introduction, you should introduce the several species of Avena, and why you wanted to study them
Table 1: Pathogen DNA. You should use consistent names for the genotypes, like genus, species, common name, variety, etc. The origin of these genotypes should be reported somewhere in the manuscript. Tables 1 and 2 could be merged to have an overall view.
L323: Fusarium in italics
L349: artificial inoculation (check here and across the manuscript)
L350-354: this approach of growing replicates of the same experiment in two successive steps is not correct!
L370: put 6 in superscript
L373: this method should be better described to be reproducible by other researchers.
L381: how did you collect the mycelium?
L382-384: re-write. It is not clear what is about plants and fungi.
L399: don't use greek letters for ng
L407: A. sativa. Check abbreviation of Latin names after the first occurrence

Author Response

First of all, I would like to thank very much the reviewer for his time and energy and for very useful comments improving the quality of the manuscript. All the comments and remarks were accepted and supplanted to the text.

Point 1: The research is interesting, well-conducted, and well-written. However, I have one major concern about the experimental design: the three replicates of the experiment were not established at the same time, i.e. two replicates (inoculated plants) were grown in a first step, and another replicate (non-inoculated plants) were grown later in a second step. This is not correct, because the plants were not cultivated in parallel. You even mentioned that B-glucan content may be affected by the environment (L255-257). I believe that the research is worth publishing with minor revisions, but frankly speaking, this concern is problematic for stating about sound results, and I think that it cannot be solved now.

Response 1: The text in M and M was written incorrect. For sure, the design of the experiment was made that way, that all the plants used in the experiment were grown at the same time and under the same conditions. That means, that control plants and artificially inoculated plants were grown together. The design of the project was corrected in the text (L327-340).

Point 2: L56: ZEN

Response 2: The abbreviation was corrected in the text (L50, L58).

Point 3: L74-77: this is clear to me, however, I believe that it should be re-written to be more clear and easy-to-read. I would formulate a scientific hypothesis to be tested. Also, at L334 you mentioned that genotypes were selected upon the B-glucans content. Indeed, it is not clear if B-glucan content is previously known information or it was determined in this work. Please clarify in M&Ms and the description of the aim of this article.

Response 3: The aim of the work was re-written in the text of abstract, introduction and conclusions to make the aim of the work clearer. Also scientific hypothesis was supplemented (L20-21, L78-82).  

Point 4: L 117: the term "pathogenic DNA" is not correct! You might want to write "pathogen DNA" or "Fusarium DNA" or "fungal DNA" or FC DNA", etc.

Response 4: In the whole text the term “pathogen” DNA was used instead to “pathogenic”.

Point 5: L76: you could cite these fungal species earlier in the text when the toxins are described. In the Introduction, you should introduce the several species of Avena, and why you wanted to study them

Response 5: Fungal species F. culmorum and F. graminearum are cited in the text of Introduction (42-43). Also “wild” species of Avena are mentioned in the text of introduction (L76-78).

Point 6: Table 1: Pathogen DNA. You should use consistent names for the genotypes, like genus, species, common name, variety, etc. The origin of these genotypes should be reported somewhere in the manuscript. Tables 1 and 2 could be merged to have an overall view.

Response 6: Tables 1 and 2 were merged and the formal aspect of the new table was changed to clarify samples/genotypes used in the experiment (L99).

Point 7: L323: Fusarium in italics

Response 7: Not only in this line, but in the whole text Fusarium was modify to be written in italics.

Point 8: L349: artificial inoculation (check here and across the manuscript)

Response 8: In the whole text artificial infection was replaced by “inoculation”, also in key words

Point 9: L350-354: this approach of growing replicates of the same experiment in two successive steps is not correct!

Response 9: I am sorry for this mistake. I modify the whole chapter “Artificial inoculation” (L341-355).

Point 10: L370: put 6 in superscript

Response 10: The number 6 was rewritten, but also the whole text of the chapter 4.5 Extraction and quantification of DNA was rewritten (L381-408).

Point 11: L373: this method should be better described to be reproducible by other researchers.

Response 11: The chapter 4.4 Quantification of DON was rewritten (L366-380).

Point 12: L381: how did you collect the mycelium?

Response 12: The whole text of the chapter 4.5 Extraction and quantification of DNA was rewritten (L381-408). The way of the mycelium collecting is supplemented (L382).

Point 13: L382-384: re-write. It is not clear what is about plants and fungi.

Response 13: The whole text of the chapter 4.5 Extraction and quantification of DNA was rewritten (L381-408).

Point 14: L399: don't use greek letters for ng

Response 14: The whole text of the chapter 4.5 Extraction and quantification of DNA was rewritten (L381-408).

Point 15: L407: A. sativa. Check abbreviation of Latin names after the first occurrence

Response 15: In the L31, the abbreviation of Avena sativa is mentioned.

Reviewer 2 Report

Dear Authors,

The manuscript deals with an important subject but the quality of writing and the lack of English basic rules... of sometimes verbs or other grammar needs... make it often difficult to follow.

Several modifications on methods, tables and results are needed.

Some suggestions are written on the manuscript file attached. But check carefully there are more than that.

Author Response

Response to Reviewer 2 Comments

First of all, I would like to thank very much the reviewer for his time and energy and for useful remarks to improve the quality of the manuscript. All the remarks were accepted in the corrected version of the text.

Point 1: L15 – the

Response 1: The world “the” was changed to “their” (L15)

Point 2: L18-19 – rewrite the sentence

Response 2: The sentence was deleted from the text of abstract.

Point 3: L21 – change the world order

Response 3: The world order was changed according the reviewer´s suggestion (L21)

Point 4: L28 – Fusarium in key words

Response 4: The word Fusarium was rewritten to italics (L28)

Point 5: L38, L39

Response 5: Sentence was rewritten according to reviewer´s suggestions. (L38-40)

Point 6: L40 – rewrite the sentence with better English

Response 6: The sentence was rewritten (L41-42)

Point 7: L44 – rewrite

Response 7: The part of the sentence mentioned by the reviewer was rewritten (L46).

Point 8: L62-63 – rewrite two parts of the sentence

Response 8: To parts of the sentence were rewritten (L64-65) and the idea of the sentence was more strictly cited by the author.

Point 9: L79-89 – rewrite according to suggestions

Response 9: The whole part of results was rewritten according to reviewer´s suggestions, Latin names were rewritten in italics and “the range” was rewritten to “ranged between … and …) (L84-94).

Point 10: Rewrite the name of the table 1 and 2

Response 10: Tables 1 and 2 were merged (according to reviewer 1) and the name was rewritten according to reviewer 2 comment (L99)

Point 11: correct the term “genotype” in the table

Response 11: the table was rewritten according the formal style to make clearer what are species and what genopytes of Avena sativa (L99-100).

Point 12: L102-115 – rewrite carefully in English

Response 12: All this part of the text was carefully rewritten to make it more clear and to have a sense for the reader. (L101-112)

Point 13: L116-118 – change words order

Response 13: The words order was changed according the reviewer´s suggestions. (L113-115)

Point 14: L314 – comma is missing

Response 14: The comma was added (L309)

Point 15: L386-400 – formal changes according the suggestions of the reviewer

Response 15: The chapter “Extraction and quantification of DNA” was the whole rewritten and the suggestions of the reviewer were all accepted (L367-380).

Point 16: L421 – change word cultivating

Response 16: The word cultivating was changed to cultivation (L429).

Point 17: L432 – carefully check the writing of references

Response 17: The chapter References was carefully checked and treated according to the instructions of the Journal

Reviewer 3 Report

Relationship between the Content of β-D-Glucans and Infection with Fusarium Pathogens in Oat (Avena sativa L.) Plants

Michaela Havrlentová, Veronika Gregusová, Svetlana Šliková, Peter Nemeček, Martina  Hudcovicová, Dominika Kuzmová

In this article, the authors made a scientific paper about the content of β-D-glucans in oat grain and infection with Fusarium pathogens. Twenty-two samples of different species and genotypes of oats were used in the experiment. In naked oats, the average β-D-glucans content in control samples was 3.25% and it decreased about 42% after the inoculation with F. culmorum and about 51% after the inoculation with F. graminearum. In hulled oats, β-D-glucans content in control samples was in the average 2.15% and after the inoculation it decreased about 38% (F. culmorum) and 40% (F. graminearum), respectively.

The article is well organized in the different sections. The way they describe the relationship is adequate and the data seem appropriate to me that it is in tables.

In my opinion, the article can be accepted as is.

Round 2

Reviewer 1 Report

Only two minor changes:

L31: delete (hereafter A. sativa)
L79: wild species of Avena genus